# High-Sensitivity Liquid Dielectric Characterization Differential Sensor by 1-Bit Coding DGS

**DOI:** 10.3390/s23010372

**Published:** 2022-12-29

**Authors:** Bingfang Xie, Zhiqiang Gao, Cong Wang, Luqman Ali, Azeem Muhammad, Fanyi Meng, Cheng Qian, Xumin Ding, Kishor Kumar Adhikari, Qun Wu

**Affiliations:** 1School of Astronautics, Harbin Institute of Technology, Harbin 150006, China; 2School of Electronics and Information Engineering, Harbin Institute of Technology, Harbin 150006, China; 3Ocean College, Zhejiang University, Hangzhou 310027, China; 4School of Instrumentation Science and Engineering, Harbin Institute of Technology, Harbin 150006, China

**Keywords:** liquid dielectric characterization, microwave differential sensor, defected ground structure, adaptive genetic algorithm

## Abstract

This paper presents two devices to detect the liquid dielectric characterization. The differential method was used to enhance the robustness and reduce tolerance. A basic sensor based on defected ground structure (DGS) was designed and the optimization for the squares of the DGS via adaptive genetic algorithm was applied to enhance the performance of the microwave sensor, which was shown by the difference of the two resonant frequencies. Furthermore, the electric field distribution was enhanced. Glass microcapillary tubes were used to hold samples to provide an environment of non-invasive. The optimized device exhibited the sensitivity of 0.076, which is more than 1.52 times than the basic structure. It could be considered a sensitive and robust sensor with quick response time for liquid dielectric characterization.

## 1. Introduction

Sensors have attracted a growing amount of interest because of their wide use in various areas, especially in chemical [1] and biomedical applications [2]. Sensors of different kinds have already been reported in previous literature [3,4,5]. Among all kinds of these sensors, microwave sensors are one of the most widely used sensors [6,7,8,9,10]. In particular, non-invasive sensors for industrial applications are in an increasing need because they can work without physical contact and respond rapidly since they don’t require markers or labels [11,12,13,14,15].

Microwave sensors for liquid dielectric characterization often use resonant techniques based on microwave resonance set-up in the standing wave formed from the propagating waves, which can provide higher precision and sensitivity compared with non-resonant techniques based on microwave propagation [11,12]. By etching the metal ground of the microwave structure, the defected ground structure (DGS) can be obtained, which can import a specific resonance unit. The equivalent capacitive and inductive can be changed through modifying the electric distribution and current direction in order to get new coupling resonance [16].

Nevertheless, this technology is still limited in its accuracy and sensitivity. Promoting the Q-factor is a feasible solution to improve the sensitivity of the biosensors, because the sensitivity of a microwave resonant sensor could be simply approximated as the product of Q-factor and filling-factor [17]. However, with the adding of the liquids under test (LUTs), there will be a great loss insert, which may decrease the Q-factor rapidly. On the other hand, the robustness that the sensors cannot be easily affected by the environmental influences is also important in the performance indicator of the sensors. Differential measurement can be considered as an effective solution to improve the sensor robustness [18,19,20,21,22,23]. There are two sensing areas in a differential sensor, one of which is designed as a test component and the other as a reference. The difference between the measured values of the two sensing areas will be used as the output variable of the differential sensor [24,25]. The influences of environmental factors, which might cause frequency shifts at these two sensing areas, are treated as common-mode perturbations and can be suppressed by using differential measurement. In the current works of differential measurement, most of the sensors require four ports [26,27], which increases the fabrication and measurement cost and makes it inconvenient to use.

In recent years, genetic algorithms have been applied to optimize the structures of microwave devices [28], which encode the potential solutions of specific problems on simple chromosome-like data structures and apply recombination operators to these structures so as to preserve key information [29]. The genetic algorithm optimizer modeled on the concepts of natural selection and evolution is a robust random search method. It starts with a generation of random chromosomes and evaluates the structures composed of these chromosomes to assign reproductive opportunities. Those chromosomes with better solutions to the target problem have more chances to reproduce than those with poorer solutions. Therefore, when the model converges, the best result can be found. This method can avoid a trap into a locally optimal solution. However, the probability of mutation and the probability of crossover are chosen by experience and are fixed in traditional GAs [30,31], which might cause GAs to become stray and make their convergence very slow. Therefore, the adaptive genetic algorithm (AGA) is used to solve this problem.

In this paper, a differential sensor based on DGS is presented to improve the robustness of the sensors in a complex measurement environment, especially to reduce the disturbance of the environment such as the temperature change when adding samples and the noise of the wires of the VNA. After that, a 1-bit coding DGS based on AGA is proposed to optimize the performance of the basic structure. The proposed sensors are two ports, which can solve the problem of the high requirement caused by the four ports’ sensors. The coding one has a higher sensitivity when compared to the basic structure on liquid dielectric characterization. Simulation and experiment indicate that the proposed DGS differential sensor optimized by AGA can improve the performance of the microwave sensors. The optimized sensor presented in the paper is sensitive, robust, has a quick response time, and is convenient for measurement.

## 2. Materials and Methods

### 2.1. The Design of the Sensor

GA seeks the global optimal value by introducing the concept of natural genetic as the independent variable and environmental fitness as dependent variable in the function into the computer. Then the individual chromosomes start crossover, inherit, and mutation. The fitness function tends to the optimal value with gradually converging of independent variables diversity. AGA adjusts the probability of mutation (*P_m_*) and the probability of crossover (*P_c_*) in every iteration rather than chosen by experience like traditional GA to get the best fitness value with a specific sequence or matrix, as shown in Equations (1) and (2) [31,32]. By using the AGA, it could converge easier and faster to obtain a globally optimum solution, which can provide the best performance of the sensors.
(1)$$P_{c}=\left\{\begin{array}{c} \frac{k_{1}(f_{\max}-f^{\prime })}{f_{\max}-f_{avg}},f^{\prime }\geq f_{avg}\\ k_{3},\;\;\;f^{\prime }\geq f_{avg} \end{array}\right.$$
(2)$$P_{m}=\left\{\begin{array}{c} \frac{k_{2}(f_{\max}-f^{\prime })}{f_{\max}-f_{avg}},f^{\prime }\geq f_{avg}k_{1}=k_{3}=1.0\\ k_{4},f^{\prime }\geq f_{avg}k_{2}=k_{4}=0.5 \end{array}\right.$$
where *f_max_* represents the best fitness value, *f_avg_* represents the average fitness value, and *f*′ is individual fitness value.

The flowchart based on AGA is shown in Figure 1.

The presented structure in this paper is a differential structure with a two-way transmission line on the top side and a DGS on the bottom side, as shown in Figure 2a–d. This kind of structure is named as the basic structure. The square and gap of the DGS could be considered as the capacitor and inductor, respectively, which would provide the resonant at the target frequency. There will be one resonant frequency while adding the same liquid and it will be separated to two while the liquid samples are different. The relative permittivity of the substrate dielectric is *ε_r_* = 6.15. The geometrical parameters of the basic design are provided in Table 1.

The AGA is used to optimize the basic structure, after which the optimized structure is presented. In this paper, the squares of the DGS are the area to be optimized. A gene-like sequence is used as the independent variable. Thus, the length of chromosome is set as 256, and there are 20 initial populations to iterate 30 times. The design area, which is the square of the DGS with a size of 6.4 mm × 6.4 mm, is divided into a matrix of 16 × 16 with 0.4 mm × 0.4 mm for each patch. And the four squares are mirrored in order to make the two ways of the differential sensor consistently. As known from the official help file of the CST Studio Suite, CST could be fully controlled by VBA script, which is almost 100% compatible with MATLAB. Therefore, this work is established in MATLAB and accomplished modeling by sequence and extracting S-parameters through CST-MATLAB API.

The raw gene sequence is given in Equation (3) with a kind of 1-bit coding on the DGS, where 0 means the etching of the copper patch in this area, while 1 means reservation of the patch. The optimized result is shown in Figure 3a and the details are shown in Figure 3b.
(3)$$x=[x_{1},x_{2},x_{3}\ldots ,x_{n}],\;x_{i}=1,0$$

The raw gene sequence is reshaped into the matrix (16 × 16) shown in Equation (4), named as **x_reshape_**.
(4)$$x_{reshape}=\left[\begin{array}{l} x_{1,1}\;x_{1,2}\;\ldots \;x_{1,16}\\ x_{2,1}\;x_{2,2}\;\ldots \;x_{2,16}\\ \ldots \;\ldots \;\ldots \;\ldots \\ x_{16,1}\;x_{16,2}\;\ldots \;x_{16,16} \end{array}\right]$$

The major applications of the AGA were to seek the maximum fitness function. In this work the fitness function is just chosen by the difference of the two resonant frequencies after adding the LUTs, which could represent the sensitivity of the sensors directly.

Both the basic structure and the optimized one are worked between 1.8 GHz to 2.5 GHz, which is a frequently used medical band.

### 2.2. Method and Material to Measure

To characterize the sensor, with DI water as the reference sample, N, N-Dimethylformamide, methanol, acetone, and ethanol were used as the samples at 25 °C approximately.

The liquid’s relative permittivity can be written as a complex quantity *ε* = *ε*′ − j*ε*″, which for a polar liquid is well-defined by Debye model [33] and could be written as Equation (5):(5)$$\varepsilon (\omega )=\varepsilon_{\infty }+\frac{\varepsilon_{s}-\varepsilon_{\infty }}{1+j\omega \tau }$$
where *ε*′ corresponds to the real part, *ε*″ corresponds to the imaginary part, *τ* is relaxation time constant, *ε*_s_ is the static permittivity, and *ε_∞_* is the high frequency permittivity

The complex permittivity of the LUTs at around 2 GHz obtained by references [34,35] is shown in Table 2.

The proposed designs were fabricated by Teflon subtraction and the biological tests were conducted with the Vector Network Analyzer Keysight N9916A, shown in Figure 4 and Figure 5. The high-grade sodium heparin glass disposable microcapillary tubes (Hansol Science, Seoul, Republic of Korea) with 1.6 mm outer diameters, 1.2 mm inner diameters, 75 mm ± 0.5 mm lengths, and 75 µL capacities were utilized to hold the samples to provide the measurement as non-invasive.

The steps of the experiment are as following. First, the sensors are connected with the VNA and set to the horizontal. Then a little glue is used at the edge of the sensors as an immobilization and a mark, which will not affect the results of the *S*_21_. A disposable microcapillary tube with DI water is filled and fixed in one of the sensing areas as the reference sample. After that, other tubes with LUTs are filled and put into the other sensing areas orderly at the mark. The tubes are easy to fill because of the capillarity. Finally, the results are recorded and analyzed.

## 3. Results

The simulation results of the two bare devices for *S*_21_ are given in Figure 6, which shows that the structure could provide a much sharper peak and a much higher Q-factor after optimization by the AGA, which will provide a higher response after adding LUTs. The Q-factor is about 130 for the basic structure and 1235 for the optimized one in simulation. When it comes to the results for the electric field distribution, it is also obvious. The optimized structures show much higher intensity values of the cut-plane of one way in the central-flat structures, shown as Figure 7a,b, which guarantees a higher sensitivity of the microwave sensor. The sensitive area in the AGA-optimized structure is about 0.35 mm high in simulation with the electric field higher than 5000 V/m, which can cover the whole microcapillary tubes well, so all the samples used will be analyzed when the sensors are working.

The measurement results of the two bare devices are shown in Figure 8. There may be slight differences between the measurements and simulations in Figure 6. It is because there must be a 0.01 mm wide overlapping with the nearby mesh to ensure a connection between each neighboring patch, but there will be some mismatching tolerance. However, the resonant frequency and the magnitude can be acceptable. Furthermore, the Q-factor of the optimized one is 21.8, which is also higher than the basic one of 12.6. The operating frequencies of the basic sensor and the optimized one are 2.375 GHz and 2.180 GHz, respectively.

When adding different LUTs into the microcapillary tubes, the other one, which is filled with DI water, will be on one way as a reference. There will be two resonant frequencies since the liquid dielectric characterizations are different from water. The results of measuring the basic device and the optimized one are shown in Figure 9 and Figure 10, respectively. The result will be visible on the instrument’s display almost at the same time as the microcapillary tubes set on the devices, which indicates the response times for the sensors proposed are within 1 s.

## 4. Discussion

The resonance frequencies and insertion losses of each LUT of the sensors are measured. The mathematical model is evaluated by multiple regression analysis with the difference in the frequency and the magnitude of *S*_21_ at the resonance frequencies as a function of complex permittivity, which can be expressed in matrix form [36]:(6)$$\left[\begin{array}{l} \Delta f\\ \Delta S_{21} \end{array}\right]=\left[\begin{array}{cc} m_{1} & m_{2}\\ m_{3} & m_{4} \end{array}\right]\left[\begin{array}{l} \varepsilon^{\prime }\\ \varepsilon^{\prime \prime } \end{array}\right]$$
where Δ*f* is the difference of two resonance frequencies, and Δ*S*_21_ is the magnitude difference of *S*_21_.

The mathematical model used to measure the complex permittivity related to the change in (Δ*f* and Δ*S*_21_) from taking an inverse matrix (Equation (6)) can be written as equation:(7)$$\left[\begin{array}{l} \varepsilon^{\prime }\\ \varepsilon^{\prime \prime } \end{array}\right]=\left[\begin{array}{cc} M_{1} & M_{2}\\ M_{3} & M_{4} \end{array}\right]\left[\begin{array}{l} \Delta f\\ \Delta S_{21} \end{array}\right]$$

For the optimized structure, the matrix is calculated as:(8)$$\left[\begin{array}{l} \varepsilon^{\prime }\\ \varepsilon^{\prime \prime } \end{array}\right]=\left[\begin{array}{cc} -0.1941 & -6.2254\\ 0.0665 & 0.2526 \end{array}\right]\left[\begin{array}{l} \Delta f\\ \Delta S_{21} \end{array}\right]$$

The complex permittivity calculated by Equation (8) is shown in Table 3. The error is calculated by Equation (9):(9)$$error=\frac{measurement\_value-reference\_value}{reference\_value}*100\%$$

The errors of the real part and the imaginary part of the permittivity are all within 5% between the measurement results and the references, which can show that the sensors proposed in the paper can be used to detect the liquid dielectric characterization.

Analyzing the difference between the two resonant frequencies and the real part of permittivity, the sensitivity results of the sensors can be calculated. The fitting lines between the difference of resonant frequencies and the real part of permittivity are shown in Figure 11, which are close to a straight line with the correlation coefficient greater than 90%. The functions for the basic and optimized structures are demonstrated as follows:(10)$$\Delta f(\mathrm{MHz})=-1.178\varepsilon^{\prime }+251.291$$
(11)$$\Delta f(\mathrm{MHz})=-1.648\varepsilon^{\prime }+155.775$$

The sensitivity, *S*, is given by Equation (12) [21]:(12)$$S_{\varepsilon^{\prime }}=\frac{f_{2}-f_{1}}{f_{0}(\varepsilon_{r}^{\prime }-1)}*100=\frac{\Delta f}{f_{0}}*100$$
where *f*_1_ and *f*_2_ are the resonant frequencies, *f*_0_ is the working frequency while the devices are bare, and the Δ*f* is the difference of the two resonant frequencies and could be calculated by the slope of the fitting lines.

Therefore, the sensitivities of the basic sensor and the optimized one are 0.50 MHz and 0.076 respectively, which is 1.52 times higher after AGA-optimization and indicates that it is an effective way to optimize the structure of the sensor by using 1-bit coding DGS via the AGA method. Furthermore, the detected response could be measured in real time because Δ*f* could be obtained directly during the test process.

Comparing this with other work with a similar operating frequency, as shown in Table 4, this work also has a competitive sensitivity, even better than some works without differential measurement. Using a differential sensor can eliminate some low tolerance from the losses from SMA connectors or perturbance in liquid samples, which will make the measurement more accurately.

## 5. Conclusions

In this paper, two kinds of microwave differential sensor for liquid dielectric characterization detection are presented. Differential measurement was used to improve the robustness against environmental influences. Good agreements between the simulated and measured results can be observed. The Q-factor and the electric field distribution were both higher than the basic structure by using AGA to optimize the basic structure. In addition, the sensitivity was also enhanced to 0.076, higher than 0.050 of the basic structure. Since the proposed sensors are sensitive, robust and with quick response time, it could be prospectively used in the array of the sensors to form a “fingerprint maps database” of the liquids in the future to detect the sorts and concentrations of the liquids’ mixture.

## Figures and Tables

**Figure 1 sensors-23-00372-f001:**
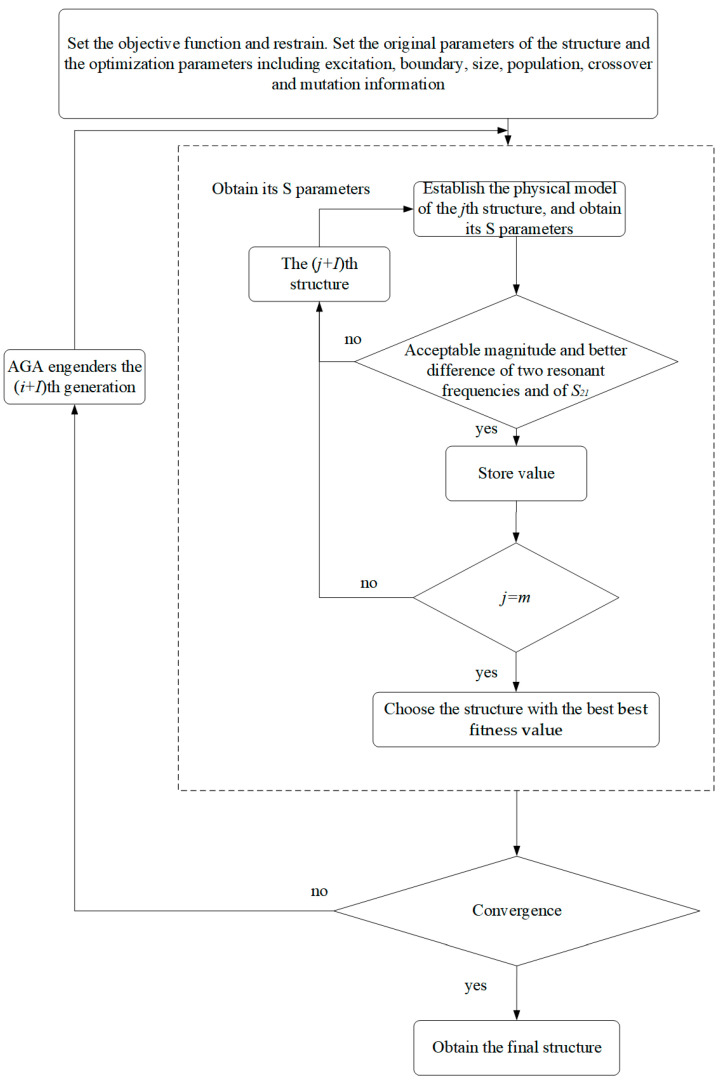
The flowchart based on AGA.

**Figure 2 sensors-23-00372-f002:**
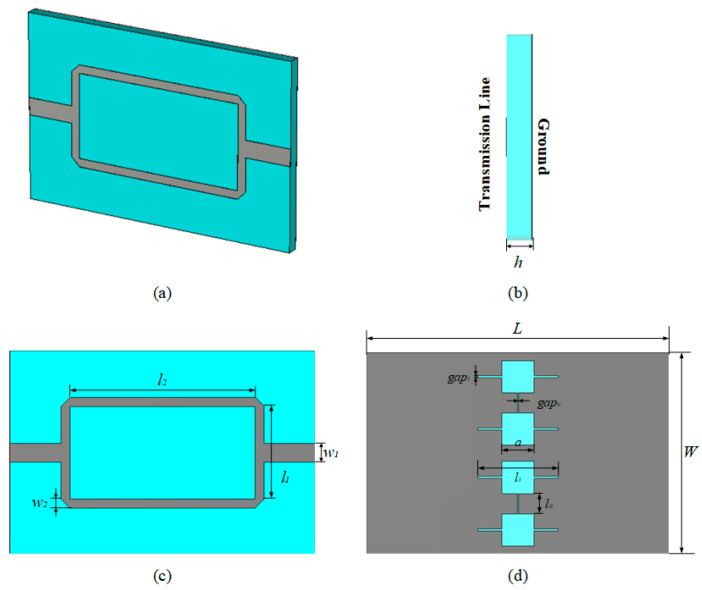
The structure of the basic sensor: (**a**) 3D structure of the basic sensor, (**b**) side view of the basic sensor, (**c**) top surface of the basic sensor, (**d**) bottom surface of the basic sensor.

**Figure 3 sensors-23-00372-f003:**
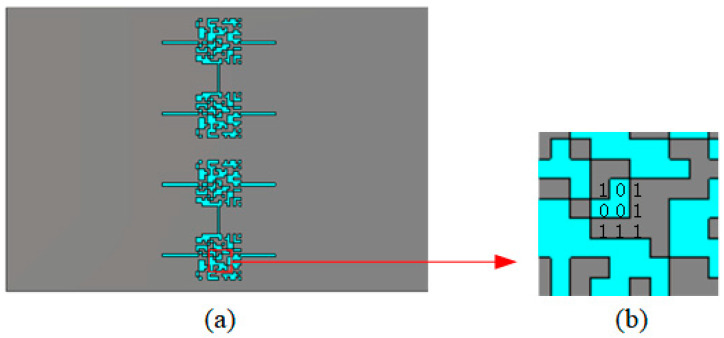
The structure of the AGA-optimized sensor: (**a**) the structure and (**b**) the details of the copper patch.

**Figure 4 sensors-23-00372-f004:**
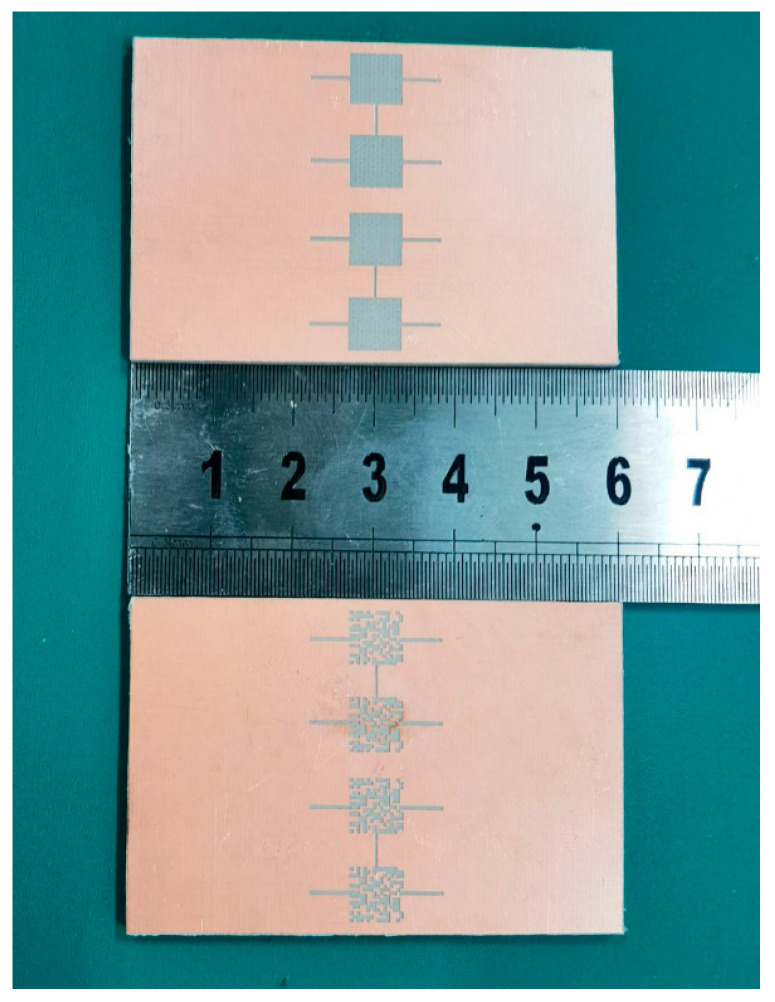
The fabricated differential microwave sensors (measurement unit: cm).

**Figure 5 sensors-23-00372-f005:**
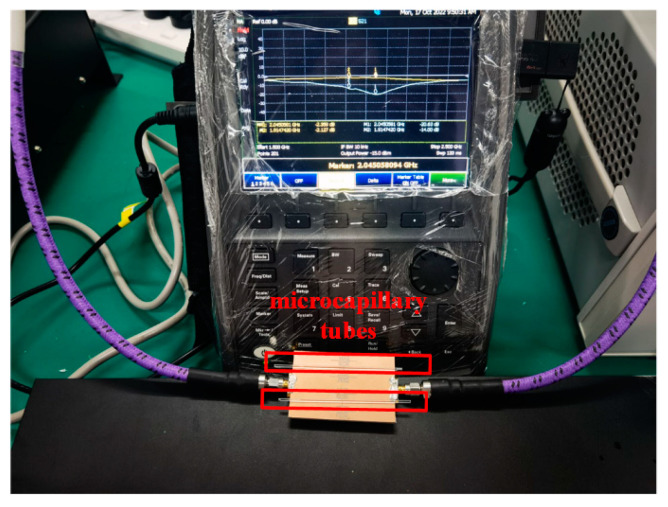
The measurement setup: optimized sensor surrounded by two microcapillary tubes and connected to the Keysight N9916A VNA. Two resonances in the *S*_21_ magnitude are visible on the instrument’s display.

**Figure 6 sensors-23-00372-f006:**
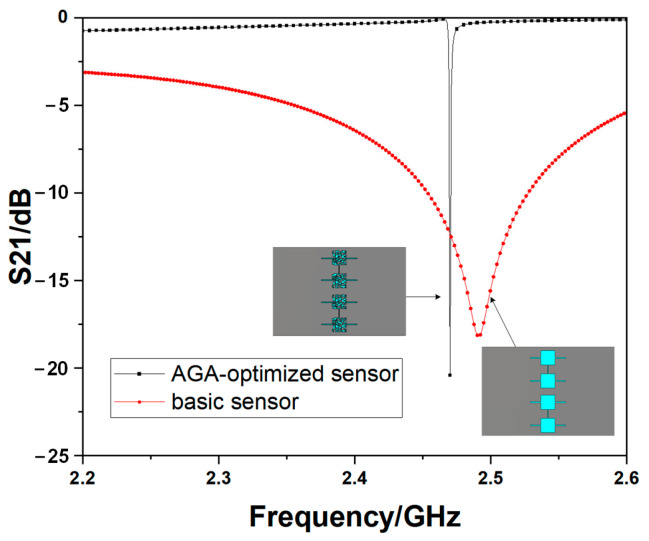
Simulation results for the two kinds of structures.

**Figure 7 sensors-23-00372-f007:**
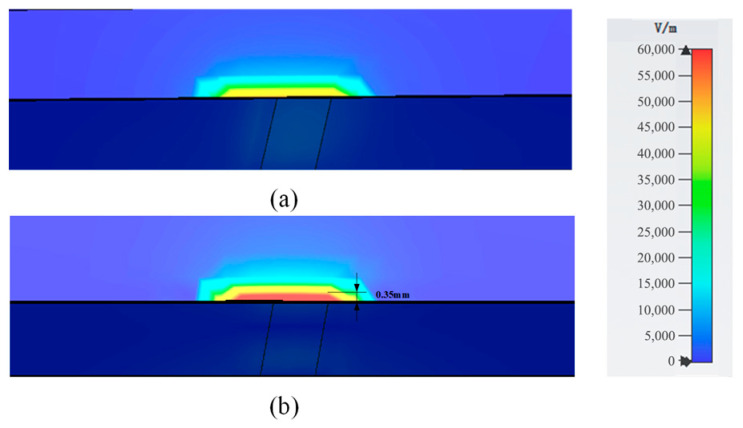
Electric field distribution of the cut-plane of one way in the central-flat structures: (**a**) the basic structure and (**b**) the AGA-optimized structure.

**Figure 8 sensors-23-00372-f008:**
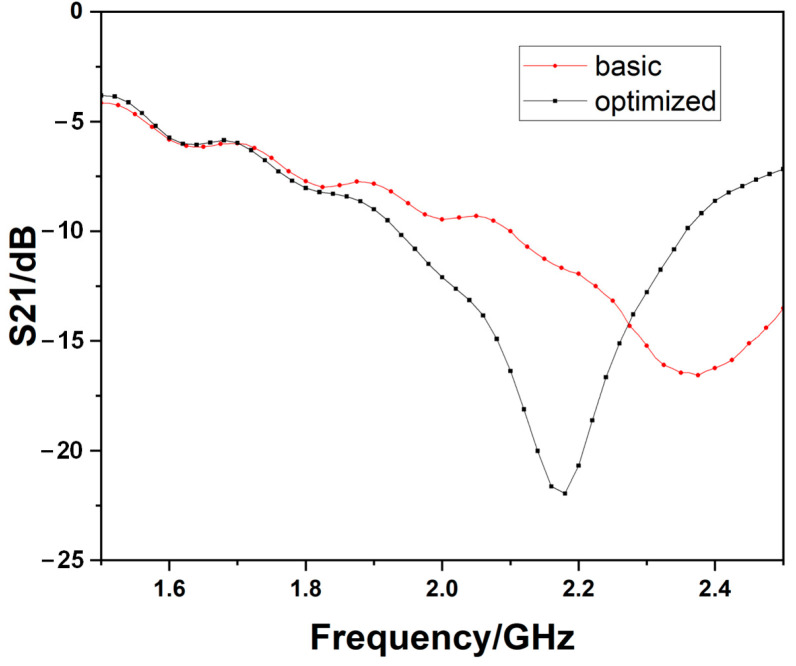
The measurement results of the bare devices.

**Figure 9 sensors-23-00372-f009:**
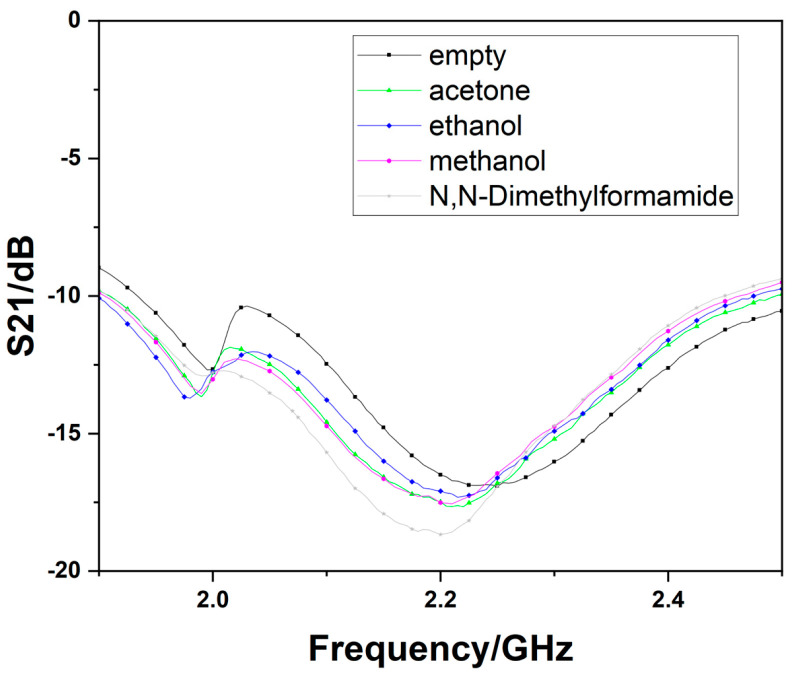
The measurement results of the basic device.

**Figure 10 sensors-23-00372-f010:**
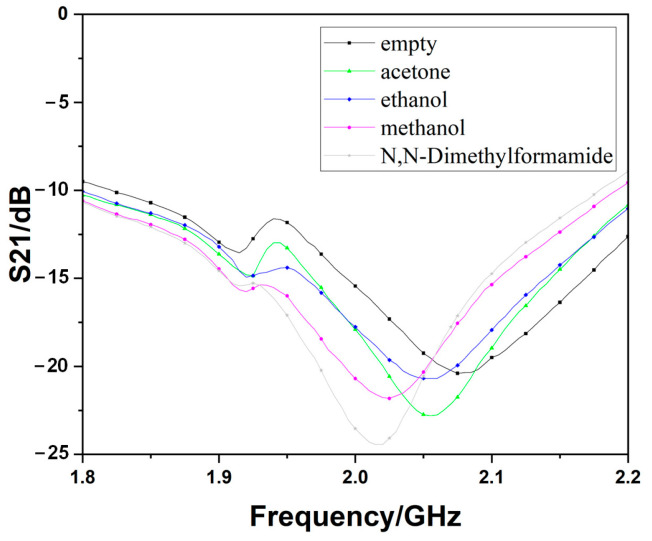
The measurement results of the optimized device.

**Figure 11 sensors-23-00372-f011:**
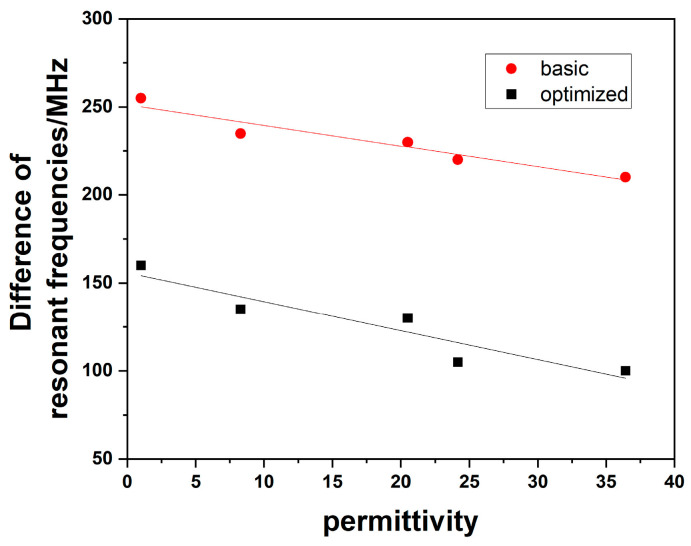
The fitting lines of the real part of permittivity for the sensors.

**Table 1 sensors-23-00372-t001:** The geometrical parameters of the basic design.

Parameters	Values (mm)	Parameters	Values (mm)
*L*	60	*l_0_*	4.0
*W*	40	*l_3_*	16
*l* _1_	18.2	*gap_0_*	0.2
*l* _2_	36.4	*gap_1_*	0.4
*w* _1_	3.7	*a*	6.4
*w* _2_	1.8	*h*	2.5

**Table 2 sensors-23-00372-t002:** The complex permittivity of the LUTs.

Liquid Samples	Complex Permittivity
N,N-Dimethylformamide	36.24 − 3.87j
methanol	24.84 − 12.28j
acetone	19.97 − 5.72j
ethanol	8.26 − 7.91j

**Table 3 sensors-23-00372-t003:** Complex permittivity obtained by measurement.

Liquid Samples	^a^ Mea of ε′	Ref of ε′	Error of ε′	Mea of ε″	Ref of ε″	Error of ε″
N,N-Dimethylformamide	36.42	36.24	0.5%	3.94	3.87	1.8%
methanol	24.15	24.84	−2.8%	11.88	12.28	−3.3%
acetone	20.50	19.97	2.6%	5.78	5.72	1.0%
ethanol	8.28	8.26	0.2%	7.93	7.91	0.3%

^a^ Mea denotes measurement.

**Table 4 sensors-23-00372-t004:** Compared results of the proposed sensor with other published sensors.

Refs.	Measurement Technique	Operating Frequency (GHz)	Differential	Sensitivity
[9]	Planar resonator	2.500	No	0.015
[10]	Minkowski-like fractal resonator	1.970	No	0.026
[37]	Bridge multiple split ring resonator	2.270	No	0.038
[38]	SIW re-entrant cavity resonators	2.637	Yes	0.037
[39]	Single stepped impedance resonator	2.450	No	0.045
**This work**	**AGA optimized DGS differential sensor**	**2.180**	**Yes**	**0.076**

## Data Availability

The data presented in this study are available in Appendix A.

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
