# Peer review of "High-Sensitivity Liquid Dielectric Characterization Differential Sensor by 1-Bit Coding DGS"

_sensors, 2022, doi:10.3390/s23010372_

Round 1

Reviewer 1 Report

This work designed two devices to detect the liquid dielectric characterization, and one was optimized via the adaptive genetic algorithm. They found that the optimized devices exhibited higher sensitivity better and electric field distribution than that of the basic structure. This work is somewhat innovative. However, certain modifications still need to be made in the method description and result display, as shown below:

1. Introduction. What are the disadvantages of current differential measurement? What improvements have been made in this article?

2.Introduction. What are the problems in applying genetic algorithms to the structure optimization of microwave devices? What improvements have been made in this article?

3. Introduction. What is the purpose of this article? What problems will this article solve?

4. Table 2, the "Dielectric characterization" in Table 2 means the real part or imaginary part of dielectric coefficient?

5. Figure 4, the basic structure and the optimized structure needs to be identified.

6. Section 2.2. It is difficult to understand the experimental process only from Figure 5 and the text description in this section. The authors should add a flow chart of experimental steps or add more text to describe the experimental steps.

7. Figure 8, the authors stated that "The measurement results of the two bare devices are shown in Fig.8, which shows good agreements between the simulated and measured results. " However, there is no simulated results in Figure 8.  Therefore, the authors should put the simulated results in Figure 8 to verify your statement.

8. Figure 9.10. The measurement results by the proposed devices should be be converted to the dielectric coefficient of these liquids, and compared to the values obtained in other literatures to verify the accuracy of the measured data in this paper.

9. Figure 11. The imaginary part of the dielectric coefficient is not seen in Figure 11. Did the author neglect the existence of the imaginary part of the dielectric coefficient in this work? If so, is this reasonable? The author should give this hypothesis and explain it in the Introduction.

10. The application scenario of the device designed in this paper should be further described to illustrate the significance of this work.

Reviewer 2 Report

I have read through the manuscript and while it presents some interesting ideas, I believe it is not yet ready for publication in its current form. There are several key areas that need further development and clarification before the manuscript can be considered for publication. In particular, the lack of a clear description of the measurement methodology and the results obtained makes it difficult to evaluate the quality of the study and its conclusions.

The authors need to provide more information about why they chose the frequency range of 1.8 GHz - 2.5 GHz for the design of the sensor. Additionally, they need to explain in greater detail how the dielectric characterization of the LUTs was carried out. The experimental setup and measurements should be described more exhaustively, including details on the position of the two microcapillary tubes, how they were arranged on the device, and how they were filled with LUTs. A photo labeled "The measurement setup" is not sufficient. The authors also need to explain how the resonant frequencies were evaluated.

Furthermore, the authors state that the measurement results of the two bare devices show good agreement between the simulated and measured results, but the simulation and measurements appear very different. In particular, the difference in terms of S21 magnitude is significant and the optimization is not evident in the fabricated device. The authors need to provide an explanation for these differences.

Overall, I believe that the manuscript would benefit from additional work before it is ready for publication.

Round 2

Reviewer 1 Report

The authors have basically solved my concerns. After carefully checking the spelling and grammar errors in the text, this paper can be accepted.

Reviewer 2 Report

The revised manuscript has significantly improved in clarity and completeness. The authors have addressed the concerns raised in my previous review, and the revised manuscript is now much easier to follow. However, there are a few additional comments that should be addressed before the manuscript can be considered for publication:

1) Please specify the measurement unit in the caption of Figure 4.

2) The Figure 5 caption is too vague. Please add more details. Is this a picture of the optimized sensor? Please report the VNA model. What is displayed on the instrument LCD panel? For example, I suggest a caption similar to: “Measurement setup: optimized sensor surrounded by two microcapillary tubes and connected to the Keysight N9916A VNA. Two resonances in the S21 magnitude are visible on the instrument's display.”

3) The authors claim: “the structure could provide a much sharper peak and a much higher Q-factor after optimization” and “the Q factor of the optimized one is also higher than the basic one”. Please report the Q-factor (measured and simulated) of the two resonators (basic and optimized).

4) The authors write: “The sensitive area in the AGA-optimized structure is about 0.35-mm-height in simulation”. How did the authors estimate this value? Which value of the electric field was used as the threshold?

5) In equation 8, the use of the capital letter “Q” may be misleading due to its association with the Q-factor. Please consider changing it.

6) In equations 6, 7, and 8, the complex permittivity is related to delta_f and delta_S21. Why did the authors not consider delta_Q, since the Q-factor is related to resonator losses and therefore to the imaginary part of the complex permittivity?

7) “The errors of the real part and the imaginary part of the permittivity are all within 5%”. Please include the formula used for the estimation of this error.

8) “Since the proposed sensors are sensitive, robust and with quick response time”. No response time value has been reported in the manuscript for the proposed sensors.

9) The authors should better highlight the novel contribution of their work in the abstract and introductory section.

10) This is not the first differential sensor prototype. I suggest including new and recent references to this kind of microwave sensor. Here are a few examples:

[*] Camli, B., Altinagac, E., Kizil, H., Torun, H., Dundar, G., & Yalcinkaya, A. D. (2020). Gold-on-glass microwave split-ring resonators with PDMS microchannels for differential measurement in microfluidic sensing. Biomicrofluidics, 14(5). https://doi.org/10.1063/5.0022767

[**] G. Gugliandolo, G. Vermiglio, G. Cutroneo, G. Campobello, G. Crupi and N. Donato, "Inkjet-Printed Capacitive Coupled Ring Resonators Aimed at the Characterization of Cell Cultures," 2022 IEEE International Symposium on Medical Measurements and Applications (MeMeA), 2022, pp. 1-5, doi: 10.1109/MeMeA54994.2022.9856582.

[**] Ebrahimi, A., Beziuk, G., Scott, J., & Ghorbani, K. (2020). Microwave differential frequency splitting sensor using magnetic-LC resonators. Sensors, 20(4), 1066. https://doi.org/10.3390/s20041066

11) There are different typos in the current version of the manuscript:

Line 68: “problem.In” missing space;

Table 1: “Values(mm)” add space: “Values (mm)”;

Table 1: use subscript for the “Parameters” column similar to Fig. 2;

line 110: “presented In this paper”. Missing “.”;

Table 2: Put the table caption in a new line;

Line 150: “Fig.5.The” missing space;

Line 199: “The resonance frequency and insertion losses”. “frequencies” or “loss”?

Lines 201 and 204: “multitude”. Do you mean “magnitude”?

Table 3. Does “Mea” stand for measurement? It is not clear at all;

Line 220: “Figure 11.,”.

Finally, sometimes the authors used “Figure” and other times “Fig.”. Please be consistent.
